# The Stack:
# 3 TB of permissively licensed source code

**Denis Kocetkov**[*]
*ServiceNow Research*

**Raymond Li**[*]
*ServiceNow Research*

**Loubna Ben Allal**[*]
*Hugging Face*

**Jia Li**
*Independent Researcher*

**Chenghao Mou**
*Independent Researcher*

**Carlos Muñoz Ferrandis**
*Hugging Face*

**Yacine Jernite**
*Hugging Face*

**Margaret Mitchell**
*Hugging Face*

**Sean Hughes**
*ServiceNow*

**Thomas Wolf**
*Hugging Face*

**Dzmitry Bahdanau**
*ServiceNow Research*

**Leandro von Werra**[‡]
*Hugging Face*

**Harm de Vries**[‡][*]
*ServiceNow Research*

## Abstract

Large Language Models (LLMs) play an ever-increasing role in the field of Artificial Intelligence (AI)–not only for natural language processing but also for code understanding and generation. To stimulate open and responsible research on LLMs for code, we introduce The Stack, a 3.1 TB dataset consisting of permissively licensed source code in 30 programming languages. We describe how we collect the full dataset, construct a permissively licensed subset, present a data governance plan, discuss limitations, and show promising results on text2code benchmarks by training 350M-parameter decoders on dif-

---

[*]Corresponding authors (denoted by ‡) can be contacted at `contact@bigcode-project.org`

ferent Python subsets. We find that (1) near-deduplicating the data significantly boosts performance across all experiments, and (2) it is possible to match previously reported HumanEval and MBPP performance using only permissively licensed data. We make the dataset available at `https://hf.co/BigCode`, provide a tool called "Am I in The Stack" (`https://hf.co/spaces/bigcode/in-the-stack`) for developers to search The Stack for copies of their code, and provide a process for code to be removed from the dataset by following the instructions at `https://www.bigcode-project.org/docs/about/the-stack/`.

## 1 Introduction

Large Language Models (LLMs) have emerged as a powerful tool for Natural Language Processing (NLP) (Brown et al., 2020; Bommasani et al., 2021; Zhang et al., 2022; Chowdhery et al., 2022; BigScience Workshop, 2022). These large transformer models are pre-trained on large internet corpora and have shown impressive zero and few-shot performance on numerous NLP tasks, often by prompting the model with a natural language description of the task at hand (Brown et al., 2020). More recently, researchers have started exploring LLMs for coding applications (Chen et al., 2021; Fried et al., 2022; Nijkamp et al., 2022). Code LLMs are trained on large collections of source code and enable the synthesis of programs from both natural language descriptions and other code snippets. Such models can assist professional developers with programming tasks, for example, by auto-completing code snippets, generating docstrings for a given function signature and body, or suggesting unit tests for a codebase.

One of the challenges faced by researchers working on code LLMs is the lack of openness and transparency around the development of these systems. While a handful of papers on code LLMs has been published, they do not always give full insight into the development process. Some research groups have made their code LLMs available through a paid API service (Chen et al., 2021) or a commercial product[1]. Other groups have published the model weights (Nijkamp et al., 2022; Fried et al., 2022) but did not release the training data. It is, therefore, difficult for researchers to fully reproduce these models because they first need to investigate the nuanced details of the data processing. For example, several groups have reported that (near) deduplication of the training data is an important preprocessing step for both natural language (Kandpal et al., 2022; Hernandez et al., 2022) and source code (Allamanis, 2019). Instead of letting research groups independently iterate over such preprocessing details, we argue that the research community would make progress faster if high-quality pre-training datasets, supported by data cards (Gebru et al., 2021; Bender & Friedman, 2018), were more broadly shared.

We believe openness around the pre-training data is also important given the ongoing legal discussions around the use of open-source code repositories for training (commercial) LLMs. Some have argued that machine learning models are a derivative work of the training material, and that therefore the resulting models and inferred outputs must comply with the terms of any licenses on the training material. This argument has been made most forcefully by advocates of copyleft licenses (Kuhn, 2022), but has also been made for permissive licenses (Butterick, 2022). Others have taken the position that code LLM developers likely benefit from copyright law exceptions, such as fair use under U.S. copyright law, when using available code repositories for model training purposes (Rothchild & Rothchild, 2022). But even if regulations currently permit the use of public source code for training ML models, it is worth discussing whether these practices align with the ethical values of society and the broader research community. For example, one could argue that code creators should have the rights to meaningfully control whether their data is included in the training set (Jernite et al., 2022). One could also have ethical concerns around the inclusion of Personally Identifiable Information (PII) and malicious code in the training data, as code LLMs might output such sensitive data (Carlini et al., 2021; Kandpal et al., 2022) or unsafe programs (Khlaaf et al., 2022) during deployment. We believe open datasets benefit from external scrutiny in addressing such data issues, as researchers can investigate the dataset and report issues directly to the maintainers.

In this paper, we take a step towards open and responsible research on code LLMs and release The Stack, a large dataset of permissively licensed source code. By releasing a dataset that can be shared, inspected, and

---

[1]For example, Microsoft's Copilot (`https://github.com/features/copilot`) and Amazon's CodeWhisperer (`https://aws.amazon.com/codewhisperer/`).

| Language | The Stack† | CodeParrot† | AlphaCode | CodeGen | PolyCoder† |
|---|---|---|---|---|---|
| Assembly | 2.36 | 0.78 | | | |
| Batchfile | 1.00 | 0.7 | | | |
| C | 222.88 | 183.83 | | 48.9 | 55 |
| C# | 128.37 | 36.83 | 38.4 | | 21 |
| C++ | 192.84 | 87.73 | 290.5 | 69.9 | 52 |
| CMake | 1.96 | 0.54 | | | |
| CSS | 145.33 | 22.67 | | | |
| Dockerfile | 1.95 | 0.71 | | | |
| FORTRAN | 3.10 | 1.62 | | | |
| GO | 118.37 | 19.28 | 19.8 | 21.4 | 15 |
| Haskell | 6.95 | 1.85 | | | |
| HTML | 746.33 | 118.12 | | | |
| Java | 271.43 | 107.7 | 113.8 | 120.3 | 41 |
| JavaScript | 486.20 | 87.82 | 88 | 24.7 | 22 |
| Julia | 3.09 | 0.29 | | | |
| Lua | 6.58 | 2.81 | 2.9 | | |
| Makefile | 5.09 | 2.92 | | | |
| Markdown | 164.61 | 23.09 | | | |
| Perl | 5.50 | 4.7 | | | |
| PHP | 183.19 | 61.41 | 64 | | 13 |
| PowerShell | 3.37 | 0.69 | | | |
| Python | 190.73 | 52.03 | 54.3 | 55.9 (217.3) | 16 |
| Ruby | 23.82 | 10.95 | 11.6 | | 4.1 |
| Rust | 40.35 | 2.68 | 2.8 | | 3.5 |
| Scala | 14.87 | 3.87 | 4.1 | | 1.8 |
| Shell | 8.69 | 3.01 | | | |
| SQL | 18.15 | 5.67 | | | |
| TeX | 4.65 | 2.15 | | | |
| TypeScript | 131.46 | 24.59 | 24.90 | | 9.20 |
| Visual Basic | 2.73 | 1.91 | | | |
| Total | 3135.95 | 872.95 | 715.1 | 314.1 | 253.6 |

Table 1: The size of The Stack (in GB) compared to other source code datasets used for pre-training LLMs. † indicates the dataset is publicly released. The Stack is more than three times the size of CodeParrot, the next-largest released code dataset.

used for pre-training, we hope to make the LLM development process more reproducible and transparent. This work describes how we collected the data from Github and present evidence that the dataset is a useful resource for developing competitive code LLMs. More specifically, we make the following contributions:

- We present The Stack, a large dataset with 3.1 TB of *permissively licensed* source code in 30 programming languages. We release this dataset along with a near-deduplicated version at `https://hf.co/BigCode`.

- We train 350M decoder-only transformers on several python subsets of the data and find that removing *near-duplicates* significantly boosts performance in all experiments. We show it is possible to reproduce text2code performance of Codex (Chen et al., 2021) and CodeGen (Nijkamp et al., 2022) by only using permissively licensed data. We outperform these models by a large margin if we train on the all-license version of the dataset.

- We acknowledge that some developers do not wish their code to be used for pre-training LLMs and, therefore, start experimenting with giving developers the possibility to have their data removed from

the dataset. We present the details of this opt-out process in a data governance plan in Section 3.2. We also provide further instructions for removal requests at `https://www.bigcode-project.org/docs/about/the-stack/`.

## 2   Related Work

**Code LLMs**   A growing body of research has trained large-scale transformer models on source code. Several groups have explored decoder-only models with a causal language modeling objective (Chen et al., 2021; Austin et al., 2021; Nijkamp et al., 2022; Christopoulou et al., 2022; Izadi et al., 2022; Xu et al., 2022) and generally found that larger models are increasingly capable of synthesizing programs from natural language descriptions. A few studies have used such decoder-only models for code-infilling tasks via a causal masking mechanism (Fried et al., 2022; Bavarian et al., 2022). Researchers have also investigated encoder masked language models (Feng et al., 2020; Kanade et al., 2020) and encoder-decoder architectures with various training objectives (Li et al., 2022; Ahmad et al., 2021; Wang et al., 2021; Roziere et al., 2021).

**Datasets for pre-training code LLMs**   GitHub has been a frequently used resource for pretraining large-scale code LLMs. Google BigQuery[2] provides a snapshot of permissively licensed repositories on GitHub and can be filtered through a SQL query. AlphaCode (Li et al., 2022), BLOOM (Laurençon et al., 2022), CodeGen (Nijkamp et al., 2022), and InCoder (Fried et al., 2022) all included this resource in their pre-training dataset. A snapshot of this GitHub data is also publicly available on the HuggingFace hub[3] as the GitHub-Code dataset under the CodeParrot project. PolyCoder (Xu et al., 2022) collected a code dataset by scraping GitHub repositories and released a catalog of their downloaded repositories and files. We compare our work against these code datasets in Table 1 and elaborate more on these differences in Section 3.3.

Another frequently used resource is the CodeSearchNet corpus (Husain et al., 2019). This corpus first collected a large number of public and permissive repositories from GitHub, which were then further processed with treesitter[4] to extract pairs of functions (methods) and docstrings. The dataset contains such pairs for six programming languages: Go, Java, JavaScript, Python, PHP and Ruby. CodeBERT (Feng et al., 2020) and CodeT5 (Wang et al., 2021) used this dataset for pre-training. CodeNet (Puri et al., 2021) is another large code dataset collected from online programming contest websites. It consists of 14M code snippets and is therefore significantly smaller than The Stack (317M permissively licensed files).

Some studies have reported results by training on non-public datasets. Codex (Chen et al., 2021) collected 179 GB of python files from 54M public GitHub repositories, but did not release the data nor disclose information regarding the licensing. CodeGen collected a private GitHub dataset of 217.3 GB of permissively licensed python files. They fine-tuned models on this dataset after pre-training on the Pile (Gao et al., 2020) and the GitHub snapshot on BigQuery. While CodeGen open-sourced the model weights, they did not release the private python dataset.

**Evaluation of code LLMs**   Code LLMs are frequently evaluated on text-to-code benchmarks, in which models are tasked with completing code snippets from natural language descriptions (Austin et al., 2021; Hendrycks et al., 2021; Chen et al., 2021; Lai et al., 2022). These benchmarks usually measure the execution accuracy of generated code against a number of unit tests. Two popular text-to-code benchmarks are HumanEval (Chen et al., 2021) and MBPP (Austin et al., 2021). Recently, these benchmarks have been extended from Python to other programming languages (Athiwaratkun et al., 2022; Cassano et al., 2022). Code LLMs have also been used to solve a variety of other tasks. CodeXGLUE (Lu et al., 2021) is a set of 14 datasets for evaluating code generation models. The tasks include code-to-code tasks like clone detection, code repair, and code translation; text-to-code tasks like code search and code generation; and code-to-text tasks like generating documentation. CoSet (Wang & Christodorescu, 2019) introduced a dataset of 84,165 programs for semantic evaluation of neural program embeddings.

---

[2]`https://cloud.google.com/blog/topics/public-datasets/github-on-bigquery-analyze-all-the-open-source-code`
[3]`https://huggingface.co/datasets/codeparrot/github-code`
[4]`https://github.com/tree-sitter/tree-sitter`

**Deduplication** Recent studies have shown that deduplicating the training set can significantly improve the performance of LLMs. Lee et al. (2021) show that training corpora for language models contain many near-duplicates, and that LLM performance improves when long repetitive substrings are removed. Hernandez et al. (2022) also found that repeating even a small portion of the training data can significantly hurt model performance, potentially due to a fraction of its capacity being consumed by data memorization. Data duplication is even more present in code datasets, since it is common practice to reuse and clone code repositories of others. Indeed, Lopes et al. (2017) observed that a large proportion of GitHub data consists of clones, resulting in a high ratio of exact and near-duplicates. Allamanis (2019) study the effect of code duplication on machine learning models and show that it can result in highly inflated performance metrics. However, many existing code LLMs (Nijkamp et al., 2022; Xu et al., 2022; Li et al., 2022) only apply exact deduplication, which leaves a large number of near-duplicates in the training data.

## 3 Dataset

We describe how we collect the code dataset and create permissively licensed and near-deduplicated subsets in Section 3.1. We present a data governance plan in Section 3.2 and further analyze the data in Section 3.3.

### 3.1 Dataset Creation

**Dataset Collection** We first collected a set of active GitHub repository names from GHArchive[5], an open-source project working on archiving and releasing the public GitHub timeline. Note that these archives only contain GitHub events, such as forking a repository and commenting on an issue, and not the code repositories. We extracted unique repository names from the event archives published between January 1st, 2015 and March 31st, 2022. This resulted in a list of 220.92M unique repository names. Next, we ran a distributed compute cluster for a couple of months to clone this list of repositories. We did not successfully download all of these repositories because some of them were deleted or made private. In the end, we successfully downloaded 137.36M repositories, resulting in a clone success rate of over 62%. All repositories were downloaded between November 2021 and June 2022.

Note that we do not store binary files as we cannot use this data for pre-training. We provide the exact list of excluded binary file extensions in Appendix C. We also do not store files larger than 1 MB except if the extension is from an approved list of programming language extensions. See Appendix D for the full list. Furthermore, we avoid storing exact file duplicates by using git hashes. All repositories contain 51.76B files, but only 5.28B of them are unique (i.e., slightly more than 10%). The uncompressed size of all *stored* files is 92.36 TB.

**License detection** We describe how we extract license information for each repository. GHArchive provides license information when the repository owner explicitly sets the code license through the web interface. We find that licenses from GHArchive are available for 26.4M repositories. For the remaining 110.9M repositories, we run the go-license-detector[6]. This detector scans the repository and returns a list of predictions with the associated file, the SPDX license identifier[7], and a confidence score. We found that the detector did not detect a license for more than 80% of the repositories. MIT and Apache 2.0 are the most frequently detected licenses for 9.6% and 2.7% of the total repositories, respectively. We present the top-20 predicted SPDX identifiers in Table 2.

**Permissive license dataset** We develop a dataset of source code with only permissive licenses, i.e., with minimal restrictions on how the software can be copied, modified, and redistributed. We first provide the list of licenses which we classified as permissive in Appendix A. Note that we intentionally exclude copyleft licenses like GPL, as this community has strongly expressed the concern of machine learning models and inferred outputs violating the terms of their licenses Kuhn (2022).

---

[5] https://gharchive.org
[6] https://github.com/src-d/go-license-detector
[7] https://spdx.org/licenses/

| SPDX identifier | Number of repos (in M) | Percentage |
|---|---|---|
| not_found | 112.51 | 81.91 |
| MIT | 13.16 | 9.58 |
| Apache-2.0 | 3.72 | 2.71 |
| BSD-3-Clause | 0.76 | 0.55 |
| error | 0.58 | 0.42 |
| GPL-3.0-only | 0.55 | 0.4 |
| GPL-3.0-or-later | 0.55 | 0.4 |
| deprecated_GPL-3.0+ | 0.55 | 0.4 |
| deprecated_GPL-3.0 | 0.55 | 0.4 |
| GPL-3.0 | 0.52 | 0.38 |
| Unlicense | 0.38 | 0.28 |
| CC0-1.0 | 0.29 | 0.21 |
| GPL-2.0-or-later | 0.28 | 0.2 |
| deprecated_GPL-2.0 | 0.28 | 0.2 |
| BSD-2-Clause | 0.24 | 0.17 |
| CC-BY-4.0 | 0.2 | 0.15 |
| CC-BY-3.0 | 0.13 | 0.1 |
| GPL-2.0 | 0.11 | 0.08 |
| MPL-2.0 | 0.1 | 0.07 |
| AGPL-3.0 | 0.09 | 0.06 |

Table 2: The top 20 detected licenses in the collected repositories. We took the highest confidence prediction per repository.

After running all our experiments, it was brought to our attention that licenses such as MPL, LGPL, and EPL were erroneously labeled as permissive when they are in fact weak copyleft licenses. We have removed these weak copyleft license files from The Stack and will release the updated version to the community. The weak copyleft-licensed data is only a small part of the overall dataset (below 0.5% for the Python subset), hence we decided to not rerun experiments as we expect findings to remain unchanged. For the new version of The Stack, we rely on the Blue Oak Council[8] to classify the set of permissive licenses - see Appendix B for the full list and the updated programming language statistics.

One challenge in compiling the permissive license dataset is that each file might be part of multiple repositories. We opt to include a file in the permissive license dataset if at least one the repositories containing the file has a permissive license. With this procedure, we keep permissively-licensed files that were copied into non-permissively licensed repositories. However, it is possible that non-permissively licensed files are part of the dataset if a developer erroneously copied a non-permissively licensed file into a permissively licensed repository. We encourage users of the dataset to report files that might have been misclassified as permissively licensed.

We mark an individual repository as permissive by using the list of license predictions as follows. We first take the highest confidence prediction for each *separate* file that the license detector flags as a potential license. We then check whether all predictions are permissive licenses (see the list in A). If so, we classify the repository as permissive. Additionally, empty files, files larger than 1 MB, and files that could not be decoded are removed from the dataset. Finally, we point out that we give developers the opportunity to have their data removed from this dataset by following the instructions at `https://www.bigcode-project.org/docs/about/the-stack/`.

**Near-deduplication**  We apply near-deduplication[9] in our pre-processing pipeline on top of exact deduplication. We largely follow the implementation[10] of Allamanis (2019) and rely on a MinHash approach Broder

---

[8]`https://blueoakcouncil.org/list`

[9]See `https://github.com/bigcode-project/bigcode-analysis/tree/main/data_analysis`

[10]`https://gist.github.com/mallamanis/ce1a3624b6d1a9ec9b6966e6b7181dcd`

| Language | All-licenses | | Permissive | | | Perm. + near-dedup | | |
|---|---|---|---|---|---|---|---|---|
| | Size (GB) | Files (M) | Size (GB) | Files (M) | Perc. | Size (GB) | Files (M) | Perc. |
| Assembly | 36.04 | 1.34 | 2.36 | 0.32 | 23.8% | 1.55 | 0.24 | 75.0% |
| Batchfile | 31.05 | 2.82 | 1.00 | 0.42 | 14.9% | 0.33 | 0.28 | 66.7% |
| C | 1461.23 | 95.57 | 222.88 | 19.88 | 20.8% | 73.21 | 10.95 | 55.1% |
| C# | 644.28 | 105.96 | 128.37 | 20.54 | 19.4% | 56.75 | 12.79 | 62.3% |
| C++ | 1106.54 | 62.72 | 192.84 | 13.54 | 21.6% | 185.60 | 7.23 | 53.4% |
| CMake | 11.25 | 3.59 | 1.96 | 0.56 | 15.8% | 0.68 | 0.25 | 44.6% |
| CSS | 1040.53 | 50.47 | 145.33 | 5.73 | 11.4% | 35.60 | 3.54 | 61.8% |
| Dockerfile | 3.89 | 3.74 | 1.95 | 1.27 | 32.6% | 0.55 | 0.65 | 51.2% |
| FORTRAN | 26.67 | 1.21 | 3.10 | 0.24 | 19.8% | 1.77 | 0.17 | 70.8% |
| GO | 271.92 | 23.34 | 118.37 | 12.08 | 51.8% | 34.87 | 6.17 | 51.1% |
| Haskell | 15.79 | 2.06 | 6.95 | 0.92 | 44.7% | 3.29 | 0.64 | 69.6% |
| HTML | 9491.23 | 267.81 | 746.33 | 32.31 | 12.1% | 279.37 | 15.55 | 48.8% |
| Java | 1311.99 | 279.16 | 271.43 | 43.01 | 15.4% | 119.19 | 26.05 | 60.1% |
| JavaScript | 5820.23 | 209.51 | 486.20 | 39.28 | 18.7% | 220.94 | 25.22 | 64.2% |
| Julia | 21.75 | 0.88 | 3.09 | 0.47 | 53.4% | 1.78 | 0.34 | 72.3% |
| Lua | 88.39 | 5.18 | 6.58 | 0.81 | 15.6% | 3.60 | 0.58 | 71.6% |
| Makefile | 39.36 | 6.34 | 5.09 | 1.09 | 17.2% | 1.69 | 0.62 | 56.9% |
| Markdown | 706.8 | 135.69 | 164.61 | 28.97 | 21.4% | 73.09 | 20.91 | 72.2% |
| Perl | 49.21 | 2.74 | 5.50 | 0.55 | 20.1% | 2.16 | 0.31 | 56.4% |
| PHP | 779.66 | 115.53 | 183.19 | 34.18 | 29.6% | 90.21 | 22.65 | 66.3% |
| PowerShell | 13.26 | 1.39 | 3.37 | 0.52 | 37.4% | 1.66 | 0.33 | 63.5% |
| Python | 737.89 | 106.91 | 190.73 | 23.58 | 22.1% | 80.38 | 15.03 | 63.7% |
| Ruby | 78.63 | 30.74 | 23.82 | 6.39 | 20.8% | 22.39 | 4.12 | 64.5% |
| Rust | 78.97 | 6.3 | 40.35 | 3.09 | 49.0% | 13.65 | 1.73 | 56.0% |
| Scala | 28.37 | 6.06 | 14.87 | 2.64 | 43.6% | 6.04 | 1.55 | 58.7% |
| Shell | 71.56 | 14.01 | 8.69 | 3.66 | 26.1% | 3.98 | 2.49 | 68.0% |
| SQL | 1438.73 | 10.2 | 18.15 | 1.27 | 12.5% | 11.47 | 1.00 | 78.7% |
| TeX | 69.4 | 4.01 | 4.65 | 0.45 | 11.2% | 3.56 | 0.38 | 84.4% |
| TypeScript | 4145.01 | 75.8 | 131.46 | 19.44 | 25.6% | 120.19 | 12.90 | 66.4% |
| Visual Basic | 28.57 | 1.97 | 2.73 | 0.2 | 10.2% | 1.20 | 0.12 | 60.0% |
| Total | 29648.2 | 1633.05 | 3135.95 | 317.41 | 19.4% | 1450.75 | 194.79 | 61.4% |

Table 3: An overview of the amount of data we collected for 30 popular programming languages. We show the size and number of files for different splits of the data: the all-license, permissive license, and permissive license with near-deduplication. The "Perc." columns show the percentage of files kept after the permissive license selection and the removal of near-duplicates, respectively.

(2000); Lee et al. (2021) to find near-duplicates. We first split the files into words/unigrams based on non-alphanumeric characters and remove files with fewer than 10 tokens. Next, we compute the MinHash with 256 permutations of all documents, and use Locality Sensitive Hashing (Har-Peled et al., 2012) to find clusters of duplicates. We include another false positive check by ensuring that each file in the original cluster is similar to (Jaccard similarity exceeding 0.85) at least one other file. We find that in the permissive license dataset, 38.6% of the files are near-duplicates of other files and are removed, they also represent 53.7% of the volume of the dataset. See Table 3 for a breakdown per programming language. We find that CMake, C, C++, and HTML subsets contain many near-duplicates (45-55%) whereas SQL, Tex, and Assembly contain relatively few near-duplicates (15-25%).

## 3.2  Data Governance

One of the goals of this project is to give developers agency over their source code and let them decide whether or not it can be used to develop and evaluate LLMs, as we acknowledge that not all developers may

wish to have their data used for that purpose. Our first step to that end was to select source code with permissive licenses, i.e. those with minimal restrictions on how the software can be copied, modified and redistributed (see Section 3.1). In addition, we are giving developers the ability to have their code removed from the dataset upon request. The process for submitting and enacting removal requests will keep evolving throughout the project as we receive feedback and build up more data governance tools. The following FAQ presents the current state of this process, as well as the planned next steps.

**How do I know my data is in The Stack?**  We have a developed a tool to help users understand whether their data is in The Stack. See `https://hf.co/spaces/bigcode/in-the-stack`.

**How can I request that my data be removed from The Stack?**  In order to request that data from your repositories be removed from The Stack, we ask that you first fill out the following form with your GitHub username and the email address associated with your git activity. After submitting the form, we will invite you to a private repository on the BigCode organization and ask you to open an issue with the topic "remove my Github repositories from The Stack". This will verify your Github username and we will mark all public repositories under your username for removal in the next dataset release cycle. The verification process is manual at the moment but we are looking into ways to fully automate it.

**What data can I request be removed from The Stack?**  Currently, you can request that we remove all public repositories under the provided username. In future work, we will expand the scope of data removal requests to address requests at a finer granularity (specific repositories, specific files) and to a greater range of contribution types (for example, based on whether a file or repository contains push events associated with your username according to GHArchive).

**Can I also prevent my data from being included in future versions of The Stack?**  The removal request form will be used to validate removal requests and remove appropriate data. Validated requests and associated code pointers will also be stored in order to ensure that the code does not appear in future versions of The Stack.

**What happens to my data once I have requested its removal?**  For as long as we are maintaining The Stack dataset, we will provide regular updates to the dataset to remove data that has been flagged since the last version. The current plan is to update the dataset every 3 months, although the schedule may change based on the volume of requests received. If we are not in a position to continue maintaining the dataset, we plan to stop distributing it in its current format and update its terms of use to limit its range of applications further, including for training new LLMs. Finally, we require that people who download the dataset agree to use the most recent allowed version in order to incorporate the removal requests.

### 3.3  Dataset Analysis

To gain more insight into the collected dataset, we first analyze the amount of data per programming language, then compare against other code datasets, and finally present a more extensive analysis on the python subset.

**Data per programming language**  We present the amount of data available for 30 programming languages in Table 3. We see that the all-license dataset contains 1.6B files (29.6 TB data). Only selecting permissively licensed files reduces the dataset to  317M files (3.1 TB), i.e. only roughly 19.4% of the files are kept. For certain programming languages (e.g. CSS, Tex, Visual Basic), we find that less than 12% of the files have a permissive license. We might be able to increase this percentage by adding more licenses to the permissive license list (see Appendix A. If we further apply near-deduplication to the permissive license dataset, we end up with 194M files (1.4 TB), a reduction of almost 40%. For example, more than half of the files are removed for CMake and HTML. Another important observation is that the the majority of the dataset are coming from a few programming languages, as can be seen in Figure 1. For the permissive license dataset, the four biggest languages–HTML (746 GB), Javascript (486 GB), Java (271 GB), and C (222 GB)–consume more than 55% of the dataset volume.

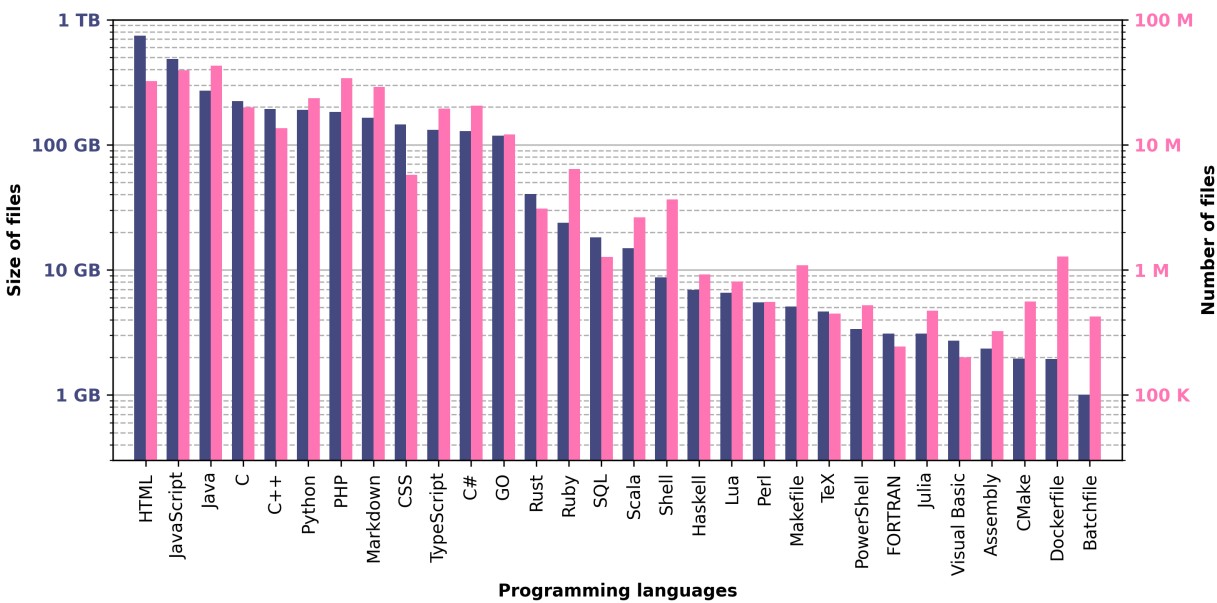

Figure 1: Histogram of the amount of data per programming language for the permissive license dataset. Note that we plot the dataset size on a log scale.

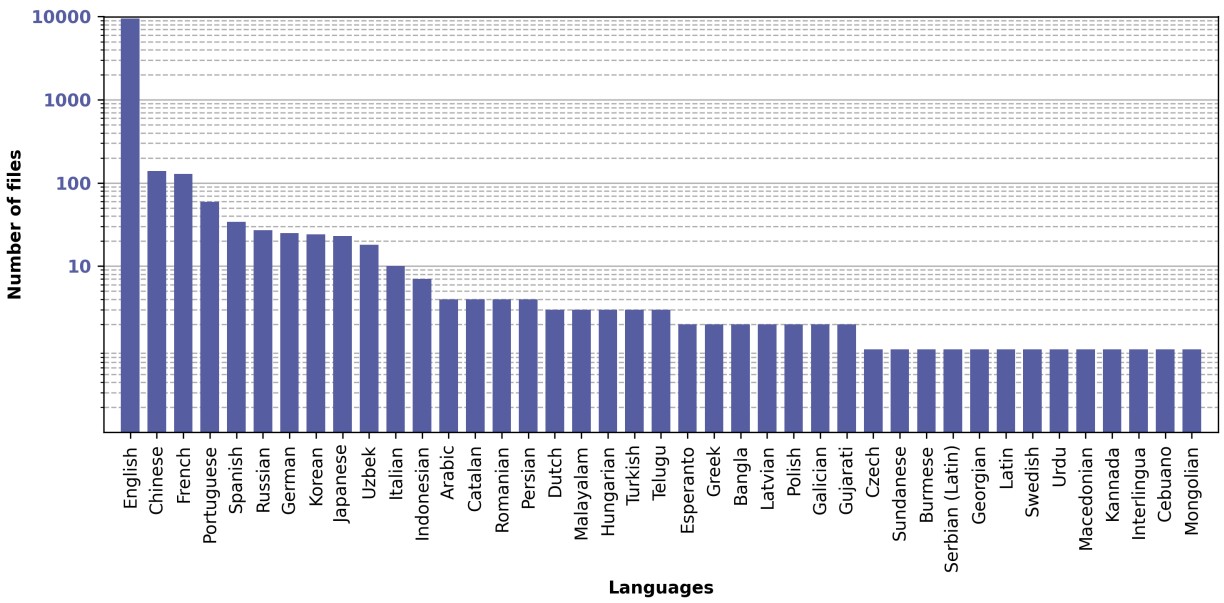

Figure 2: Histogram of natural languages in docstrings and comments from 10,000 Python files. Note that we plot the number of files on a log scale.

**Comparison with other code datasets** We compare The Stack with CodeParrot[11], AlphaCode(Li et al., 2022), CodeGen(Nijkamp et al., 2022), PolyCoder(Xu et al., 2022) in Table 1. Note that AlphaCode and CodeGen did not release their data but provided statistics on the amount of data per programming language. It is also worth mentioning that CodeParrot includes files with a copyleft license (e.g. GPL) while our permissive license dataset does not. PolyCoder does not filter for license information and also

---

[11]https://huggingface.co/datasets/codeparrot/github-code

likely contains files with copyleft license. While The Stack and CodeParrot provide source code for 30 programming languages, AlphaCode, PolyCoder, and CodeGen only provide data for 12, 12, and 6 of the languages, respectively. Furthermore, we observe that our dataset is more than 3x the size of CodeParrot, the next largest publicly available code dataset. We also see that our dataset is bigger in size than CodeParrot for each individual programming language.

**Python subset analysis** First, we investigate how many configuration and test files are in the Python subset of the permissive license dataset. These config files are abundant in GitHub repositories but do not capture the complex features of source code. To detect them, we look for mentions of keywords such as "configuration file" and "test file" in the first 5 lines. If these keywords are not present, we see if the occurrence number of either *config* or *test* literals is higher than 5% of the number of lines in the file. This filter selects 15% of the files which represent 23% of the data by volume.

Next, we estimate the number of valid Python files by using the `py_compile` module on 10,000 samples from our dataset. We find that only 0.7% of the files do not compile due to syntax errors. Specifically, we find that around 0.1% of the Python files had tokenization errors (e.g., due to inconsistent use of tabs and spaces).

Finally, we analyze the docstrings and comments in the files. This data is essential for applications such as documentation generation and natural language to code translation (Chen et al., 2021; Feng et al., 2020). We analyze a subset of 10,000 files. We first use the `ast` and `tokenize` modules to extract docstrings and comments. We find that they make up 18% of the volume of the subset and that 20% of the files had little (less than 20 characters) to no natural text.

To identify the language of the extracted docstrings and comments, we use a model from the *fasttext* library[12]. We find that 94% of the files are in English. Among the other languages, Chinese and French are popular with over 100 samples. We show the full distribution of natural languages in Figure 2. It is important to note that this language detection is imperfect since docstrings often include code examples with English keywords.

## 4 Experiments

In this work, we investigate the quality of the Python subset of The Stack. Subsequent work by the BigCode community has investigated the the quality of other subsets by training larger Code LLMs that support more programming languages (Allal et al., 2023; Li et al., 2023). Here, we train several LLMs on the Python subset of The Stack and evaluate the models on HumanEval (Chen et al., 2021) and MBPP (Austin et al., 2021). We compare against CodeGen (Nijkamp et al., 2022), InCoder (Fried et al., 2022), and Codex (Chen et al., 2021) models of similar size. We discovered late in the research process that some of the pretraining sets were contaminated with examples from the evaluation benchmarks. As running experiments is expensive and takes long, we decided to re-run only a few experiments to investigate the impact of the data contamination. In addition, we did not rerun experiments after finding out a labelling mistake in the permissive licenses (see Appendix A for full details). The results we present for permissively licensed Python data, therefore, do contain a small fraction (less than 0.5%) of weak copyleft licensed data. However, we think our main findings remain unchanged.

### 4.1 Experimental setup

**Datasets** We experiment with 4 versions of our python dataset, as well as the python subset of Code-Parrot. For our datasets we either use python files from all repositories (*all-license*) or from repositories with permissive licenses (*permissive-license*). To this data we either apply near-deduplication (*near-dedup*) or no further filtering (*none*). It is worth stressing that the near deduplication process is on top of the exact deduplication that we applied during the dataset collection. For CodeParrot we only experiment with the near-deduplicated version[13].

---

[12]See `https://github.com/bigcode-project/bigcode-analysis/tree/main/data_analysis`

[13]`https://hf.co/datasets/codeparrot/codeparrot-train-near-deduplication`

| Dataset | Filter | HumanEval | MBPP |
|---|---|---|---|
| Python-all-license | None | 338 | 1 |
| | Near-dedup | 291 | 1 |
| Python-permissive-license | None | 336 | 0 |
| | Near-dedup | 292 | 0 |
| CodeParrot | Near-dedup | 0 | 0 |

Table 4: Data contamination per training set. We report the number of files in which we find a natural language prompt of a test example.

For all datasets, we follow the filtering methods of Codex (Chen et al., 2021) and remove files with:

- an average line length greater than 100 characters

- a maximum line length above 1,000 characters

- less than 25% of the characters being alphanumeric characters

- keywords in the first few lines of the file indicating that the file was likely automatically generated

**Data contamination**  After training the models, we found that some of the training sets contained examples from the evaluation benchmarks. We detected this contamination issue by searching for exact string matches of the natural language prompts of HumanEval and MBPP. Table 4 shows how many contaminated files were found for each of the subsets. All our subsets contain HumanEval examples but only the python-all-license subsets contain MBPP examples. To investigate the impact of the data contamination, we rerun experiments on the near-deduplicated versions of the python-all-license and python-permissive-license datasets. To this end, we remove the contaminated files from these datasets. Note that we only eliminate files with exact copies of the prompts and thus do not detect paraphrases.

**Evaluation**  We evaluate models on the HumanEval (Chen et al., 2021) and MBBP (Austin et al., 2021) benchmarks. HumanEval is a text2python benchmark containing 164 programming problems. Each problem consists of a function signature, docstring, body, and some test cases that allow to verify the correctness of the generated program. Models are evaluated with the pass@$k$ metric (Chen et al., 2021): $k$ model samples are generated for each problem, and a problem is considered solved if at least one of the samples passes all the unit tests. We report the average fraction of problems solved. In practice, we sample 200 programs for each problem, and estimate pass@$k$ with the unbiased estimated proposed by Chen et al. (2021). We use nucleus sampling with top-$p = 0.95$, temperature $T \in \{0.2, 0.8\}$ and report the pass@$k$ for the best temperature.

We also evaluate on MBPP (Austin et al., 2021), a text2python benchmark consisting of crowdsourced programming problems. Each problem consists of a short natural language description and 3 unit tests. We evaluate on the test set of 500 examples. While Austin et al. (2021) include all three unit tests in the prompt, Fried et al. (2022) only include a single unit test because they observed that smaller language models (i.e., with a few billion parameters) did not benefit from more unit tests. We follow Fried et al. (2022) and only add the first unit test to the natural language prompt. Besides pass@1, we also evaluate pass@10 and pass@100 by sampling 200 programs for each problem. Similarly to HumanEval, we use nucleus sampling with top-$p = 0.95$, temperature $T \in \{0.2, 0.8\}$ and report the scores for the best temperature.

**Training details**  We experiment with decoder-only transformers trained via a causal language modeling objective. We opt for a 350M parameter model with 24 layers, a hidden dimension of 1024, 16 attention heads, and a sequence length of 2048. The model is trained for 300K iterations with a global batch size of 384 using Adam (Kingma & Ba, 2015) with $\beta_1 = 0.9$, $\beta_2 = 0.95$, $\epsilon = 10^{-8}$ and a weight decay of 0.1. The learning rate set to $3 \times 10^{-4}$ is warmed up for 175 steps, then follows a cosine decay. The model processes 235.9B tokens during training. The Byte-Pair Encoding tokenizer was trained on a 50-50 mixture of the

| Dataset | Filtering | Pass@1 | Pass@10 | Pass@100 |
|---|---|---|---|---|
| Codex (300M) | | 13.17 | 20.17 | 36.27 |
| CodeGen (350M) | | 12.76 | 23.11 | 35.19 |
| Python all-license | None | 13.11 | 21.77 | 36.67 |
| | Near-dedup | 16.60 | 27.82 | 44.00 |
| | + Decontamination | 17.34 | 27.64 | 45.52 |
| Python permissive-license | None | 10.99 | 15.94 | 27.21 |
| | Near-dedup | 13.94 | 22.36 | 37.00 |
| | + Decontamination | 12.89 | 22.26 | 36.01 |
| CodeParrot | Near-dedup | 11.23 | 18.16 | 30.37 |

Table 5: HumanEval performance of a 350M model on different training sets.

| Model | Filtering | Pass@1 | Pass@10 | Pass@100 |
|---|---|---|---|---|
| InCoder-1B | | 9.36 | 23.37 | 45.80 |
| CodeGen (350M) | | 14.09 | 30.07 | 51.80 |
| Python all-license | None | 17.41 | 33.09 | 53.59 |
| | Near-dedup | 22.99 | 39.62 | 61.00 |
| | + Decontamination | 21.82 | 37.55 | 58.28 |
| Python permissive-license | None | 11.60 | 23.13 | 44.99 |
| | Near-dedup | 15.94 | 31.70 | 54.69 |
| CodeParrot | Near-dedup | 6.31 | 21.50 | 45.44 |

Table 6: MBPP performance of a 350M model on different training sets.

Pile (Gao et al., 2020) and Python files from The Stack. We use a fork[14] of Megatron-LM (Shoeybi et al., 2019) for training.

## 4.2 Discussion of results

We report the HumanEval and MBPP results in Table 5 and 6, respectively.

**Near-deduplication improves performance** Applying near-deduplication to the training data yields a dramatic improvement, on both the all-license and permissive-license datasets. On the permissive-license dataset, near-deduplication improves HumanEval performance from 27.21% to 37.00% pass@100 and MBPP performance from 44.99% to 54.69% pass@100. We see a similar boost in results for the all-license dataset, where HumanEval performance improves from 36.67% to 44.00% pass@100 and MBPP performance from 53.59% to 61.00% pass@100.

**Reproducing performance with permissive licenses** We are able to reproduce text2python results of previous work with only permissively licensed source code. On HumanEval, we observe that, without near-deduplication, the performance of the permissive license dataset (27.21% pass@100) is significantly below Codex (36.27% pass@100) and CodeGen(35.19% pass@100). However, we match Codex and CodeGen performance after applying near-deduplication (37.00% pass@100). On MBPP, we observe similar findings. Without near-deduplication, the performance of the python-permissive dataset (44.99% pass@100) is significantly below CodeGen (51.80% pass@100). However, the near-deduplicated version (54.69% pass@100) surpasses the CodeGen results.

**Comparison to CodeParrot** We see that training on CodeParrot—another released code dataset— achieves 30.37% pass@100 on HumanEval, which is significantly below the performance of our released

---

[14]https://github.com/bigcode-project/Megatron-LM

dataset (37.00% pass@100). On MBPP, CodeParrot also underperforms our released dataset (45.44% vs 54.69% pass@100). CodeParrot consists of data extracted from BigQuery and is not enough to obtain a competitive model on HumanEval and MBPP.

**Impact of data contamination** Surprisingly, we find that removing contaminated files has very little impact on the text2python results. On HumanEval, there is very small drop for the permissive-license dataset (37.00% vs 36.01% pass@100), while there is a small gain for the all-license dataset (44.00% vs 45.52% pass@100). We also see a small impact for the all-license dataset on MBPP (61.00% vs 58.28% pass@100). We speculate that the impact of data contamination was minimal because (1) small models are unlikely to memorize training data and (2) there were few contaminated files.

## 5   Conclusion and Future Work

We introduce The Stack, a large dataset of more than 3 TB of permissively licensed source code. This paper described the details of the dataset collection, presented a brief dataset analysis, and showed promising results on the HumanEval benchmark. Our experimental results show that near-deduplication is an important pre-processing step for achieving competitive results on text2code benchmarks. We release all permissively licensed files for 30 common programming languages, along with a near-deduplicated version. In future work, we would like to further improve the released dataset. We are open to releasing data of other programming languages, plan to work on methods for removing PII and malicious code, and start experimenting with giving developers the possibility to have their data removed from the dataset. We hope The Stack will be a useful resource for open and responsible research on Code LLMs.

**Acknowledgement** We thank Christopher Akiki, Evgenii Zheltonozhskii, Sebastien Paquet, Torsten Scholak, and Luis Villa for their feedback on an early draft of this paper. We also thank Fanny Rancourt and Masoud Hashemi for help with the data cards. Lastly, we are grateful to ServiceNow and Hugging Face for the provided compute resources.

## 6 Limitations

**Social Impact of the Dataset**  The Stack is an output of the BigCode Project[15]. BigCode aims to be responsible by design and by default. The project is conducted in the spirit of Open Science, focused on the responsible development of LLMs for code.

With the release of The Stack, we aim to increase access, reproducibility, and transparency of code LLMs in the research community. Work to de-risk and improve on the implementation of ethical best practices of code LLMs is conducted in various BigCode working groups. The Legal, Ethics, and Governance working group has explored topics such as licensing (including copyleft and the intended use of permissively licensed code), attribution of generated code to original code, rights to restrict processing, the inclusion of Personally Identifiable Information (PII), and risks of malicious code, among other topics. This work is ongoing as of October 25th, 2022.

We expect code LLMs to enable people from diverse backgrounds to write higher quality code and develop low-code applications. Mission-critical software could become easier to maintain as professional developers are guided by code-generating systems on how to write more robust and efficient code. While the social impact is intended to be positive, the increased accessibility of code LLMs comes with certain risks such as over-reliance on the generated code and long-term effects on the software development job market.

We refer the reader to Section 7 of Chen et al. (2021) for a broader impact analysis of Code LLMs, as well as Khlaaf et al. (2022) for an in-depth risk assessment and hazard analysis of this emerging technology.

**Biases of the Dataset**  The code collected from GitHub does not contain demographic information or proxy information about the demographics. However, it is not without risks, as the comments within the code may contain harmful or offensive language, which could be learned from the models.

Widely adopted programming languages like C and Javascript are overrepresented compared to niche programming languages like Julia and Scala. We found that some programming languages such as SQL, Batch-file, TypeScript are less likely to be permissively licensed (4% vs the average 10%). This may result in a biased representation of those languages. Permissively licensed files also tend to be longer.

We also found that the English language is over represented in the docstrings and comments, as it makes up 96% of the data for Python files.

**HTML not WCAG-compliant**  One of the current limitations of The Stack is that scraped HTML for websites may not be compliant with Web Content Accessibility Guidelines (WCAG). This could have an impact on HTML-generated code that may introduce web accessibility issues.

**Personally Identifiable Information**  We noted that Personally Identifiable Information (PII) such as names and email addresses are contained in the data. This PII is already exposed to the public on GitHub, however, we do plan to remove PII in future work.

**Malicious code**  In the context of cyber-security, there is risk that large language models trained on datasets containing ransomware, malware, or other malicious code, could be used to build harmful applications. The ability to spawn new harmful applications by prompting the model using known indicators of compromise (IOCs) could reduce the experience needed for hackers to learn these techniques[16] and increase the risk of Ransomware-as-a-Service kits being developed and distributed, not on the Dark Web, but in the general public domain by citizen-hackers and activists. While more advantaged companies and communities may have resources to manage these new threats, the negative social impact of ransomware on disadvantaged communities and society in general, along with the potential legal consequences for companies that pay ransom to bad actors, is something that requires more consideration.

---

[15]https://www.bigcode-project.org
[16]See this article https://www.trendmicro.com/en_us/research/22/a/codex-exposed-helping-hackers-in-training.html.

A handful of repositories were removed because they triggered an internal security tool during the downloading phase. However, we did not fully scan the dataset for malware and, as such, warn future users of the potential for malicious code bases in The Stack. Please report your findings of malicious code to `contact@bigcode-project.org` so that it can be removed in a future release.

**Data licensing**  While we did our best to filter for permissively licensed source code, it is possible that the license detector incorrectly classified a number of repositories. Please reach out to `contact@bigcode-project.org` in case you encounter files that do not have a permissive license. We also stress that The Stack is a compilation of source files—including attribution and license notices—in the form of a dataset. Each source file in The Stack carries its own permissive license which should be respected by users of the dataset.

**Model limitations**  We show promising text2code results for smaller models on the python dataset. More research is necessary to find out whether strong results can be obtained for larger models and other programming languages than Python.

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

## A  Permissive Licenses

For the experiments presented in the paper, we use the following SPDX identifiers for our permissive license dataset:

- MIT-0

- MIT

- MIT-feh

- Apache-2.0

- BSD-3-Clause

- BSD-3-Clause-Clear

- BSD-3-Clause-No-Nuclear-License-2014

- BSD-2-Clause

- CC0-1.0

- EPL-1.0

- MPL-2.0

- Unlicense

- ISC

- Artistic-2.0

- deprecated_LGPL-3.0+

- deprecated_LGPL-2.1+

- ECL-2.0

- SHL-0.51

- MPL-2.0-no-copyleft-exception

After running all experiments, it was brought to our attention that licenses such as MPL, LGPL, and EPL were erroneously labeled as permissive when they are in fact weak copyleft licenses. We have removed these weak copyleft license files and will release the updated version of The Stack. The weak copyleft-licensed data is only a small part of the overall dataset (below 0.5% for the Python subset), hence we expect the experimental findings of the paper to remain unchanged. In the next section, we describe how we updated The Stack.

## B  The Stack v1.1

For the Stack v1.1, we rely on the Blue Oak Council[17] to classify the licenses. The new classification process results in 193 permissive licenses (which we list, for completeness, at the end of this section). The Stack v1.1 also includes more programming languages. We used the following list of programming language extensions `https://gist.github.com/ppisarczyk/43962d06686722d26d176fad46879d41` and obtained data for 370 programming languages. We show the updated statistics for the 30 popular programming languages in Table 7. In general, we find that the amount of data increased for most programming languages. Specifically, we see a large increase in data for C# (128.37 GB vs 215.07 GB), Tex (4.65 GB vs 8.19 GB), and Markdown (164.61 GB vs 245.26 GB). On the other hand, we see a minor decrease in data for Go (118.37 GB vs 112.86 GB) and Rust (40.35 GB vs 39.85 GB). We release this updated version to the research community.

- MIT

- Apache-2.0

- BSD-3-Clause

- Unlicense

- CC0-1.0

- BSD-2-Clause

---

[17]`https://blueoakcouncil.org/list`

| | All-licenses | | Permissive | | Perm. + near-dedup | |
| Language | Size (GB) | Files (M) | Size (GB) | Files (M) | Size (GB) | Files (M) |
| --- | --- | --- | --- | --- | --- | --- |
| Assembly | 36.04 | 1.34 | 2.57 | 0.36 | 1.65 | 0.26 |
| Batchfile | 31.05 | 2.82 | 1.06 | 0.44 | 0.33 | 0.28 |
| C | 1461.23 | 95.57 | 255.29 | 21.38 | 75.93 | 11.21 |
| C++ | 644.28 | 105.96 | 215.07 | 14.82 | 65.97 | 7.60 |
| C# | 1106.54 | 62.72 | 133.55 | 21.7 | 57.98 | 13.28 |
| CMake | 11.25 | 3.59 | 2.1 | 0.59 | 0.68 | 0.25 |
| CSS | 1040.53 | 50.47 | 150.2 | 5.89 | 34.86 | 3.59 |
| Dockerfile | 3.89 | 3.74 | 1.9 | 1.27 | 0.52 | 0.64 |
| FORTRAN | 26.67 | 1.21 | 3.81 | 0.29 | 2.08 | 0.19 |
| GO | 271.92 | 23.34 | 112.86 | 11.65 | 32.01 | 5.89 |
| Haskell | 15.79 | 2.06 | 5.85 | 0.8 | 2.75 | 0.58 |
| HTML | 9491.23 | 267.81 | 812.73 | 35.6 | 291.46 | 16.60 |
| Java | 1311.99 | 279.16 | 266.41 | 42.43 | 112.82 | 25.12 |
| Javascript | 5820.23 | 209.51 | 496.22 | 40.11 | 166.24 | 25.43 |
| Julia | 21.75 | 0.88 | 3.4 | 0.48 | 1.75 | 0.33 |
| Lua | 88.39 | 5.18 | 7.09 | 0.93 | 3.77 | 0.64 |
| Makefile | 39.36 | 6.34 | 6.88 | 1.48 | 2.14 | 0.80 |
| Markdown | 706.8 | 135.69 | 245.26 | 40.75 | 95.84 | 25.66 |
| Perl | 49.21 | 2.74 | 7.08 | 0.83 | 2.99 | 0.48 |
| PHP | 779.66 | 115.53 | 185.79 | 34.85 | 89.46 | 22.63 |
| PowerShell | 13.26 | 1.39 | 3.42 | 0.53 | 1.65 | 0.33 |
| Python | 737.89 | 106.91 | 200.93 | 24.21 | 80.13 | 15.15 |
| Ruby | 78.63 | 30.74 | 25.95 | 7.21 | 9.78 | 4.46 |
| Rust | 78.97 | 6.3 | 39.85 | 3.06 | 12.92 | 1.68 |
| Scala | 28.37 | 6.06 | 15.56 | 2.79 | 6.06 | 1.61 |
| Shell | 71.56 | 14.01 | 9.07 | 3.77 | 4.07 | 2.54 |
| SQL | 1438.73 | 10.2 | 19.94 | 1.39 | 12.68 | 1.07 |
| TeX | 69.4 | 4.01 | 8.19 | 0.71 | 5.86 | 0.59 |
| Typescript | 4145.01 | 75.8 | 131.01 | 19.59 | 36.61 | 12.82 |
| Visual Basic | 28.57 | 1.97 | 3.81 | 0.4 | 1.71 | 0.19 |
| Total | 29648.2 | 1633.05 | 3372.85 | 340.31 | 1212.7 | 201.90 |

Table 7: An overview of the amount of data we collected for The Stack v1.1. We show the size and number of files for different splits of the data: the all-license, permissive license, and permissive license with near-deduplication.

- CC-BY-4.0

- CC-BY-3.0

- 0BSD

- RSA-MD

- WTFPL

- MIT-0

- ISC

- ADSL

- BSL-1.0

- Zlib

- Artistic-2.0

- FTL

- MS-PL

- BSD-2-Clause-FreeBSD

- FSFAP

- BSD-Source-Code

- Apache-1.1

- BSD-4-Clause

- Ruby

- Artistic-1.0

- MulanPSL-1.0

- BSD-1-Clause

- X11

- CNRI-Python

- Beerware

- Condor-1.1

- PostgreSQL

- CECILL-B

- Intel

- Vim

- Naumen

- OML

- BSD-3-Clause-Clear

- AML

- PHP-3.01

- OpenSSL

- PSF-2.0

- Xnet

- Linux-OpenIB

- BSD-3-Clause-LBNL

- UPL-1.0

- AFL-3.0

- BlueOak-1.0.0
- Info-ZIP
- BSD-4-Clause-UC
- AAL
- LPPL-1.3c
- bzip2-1.0.6
- W3C
- W3C-20150513
- AFL-1.1
- DOC
- ICU
- CC-BY-2.0
- curl
- MTLL
- OLDAP-2.2.1
- ECL-2.0
- Adobe-Glyph
- CNRI-Python-GPL-Compatible
- BSD-2-Clause-Patent
- IJG
- PHP-3.0
- ZPL-2.1
- MIT-advertising
- NCSA
- Fair
- BSD-3-Clause-Attribution
- OLDAP-2.3
- NLPL
- BSD-3-Clause-Open-MPI
- ClArtistic
- Python-2.0
- NASA-1.3
- TCL

- Artistic-1.0-Perl

- blessing

- BSD-3-Clause-No-Nuclear-Warranty

- ImageMagick

- Net-SNMP

- Artistic-1.0-cl8

- OLDAP-2.5

- MIT-feh

- OLDAP-2.4

- MITNFA

- AFL-2.1

- libpng-2.0

- EFL-2.0

- OLDAP-2.7

- IBM-pibs

- libtiff

- OLDAP-2.8

- Cube

- Adobe-2006

- BSD-2-Clause-NetBSD

- zlib-acknowledgement

- OLDAP-2.6

- BSD-3-Clause-No-Nuclear-License-2014

- OLDAP-1.4

- Libpng

- MIT-CMU

- AFL-2.0

- JasPer-2.0

- LPL-1.02

- Zend-2.0

- TCP-wrappers

- XFree86-1.1

- FSFUL

- OLDAP-1.3
- SGI-B-2.0
- NetCDF
- CNRI-Jython
- Zed
- ZPL-2.0
- AFL-1.2
- Apache-1.0
- CC-BY-1.0
- OLDAP-2.1
- OLDAP-1.2
- OLDAP-2.0
- NTP
- LPL-1.0
- AMPAS
- Barr
- mpich2
- ANTLR-PD
- Xerox
- Spencer-94
- AMDPLPA
- BSD-3-Clause-No-Nuclear-License
- HPND
- ECL-1.0
- MirOS
- Qhull
- ZPL-1.1
- TU-Berlin-2.0
- Spencer-86
- SMLNJ
- xinetd
- OLDAP-2.2.2
- OGTSL

- MIT-enna

- Font-exception-2.0

- FSFULLR

- TU-Berlin-1.0

- xpp

- NRL

- W3C-19980720

- EFL-1.0

- eGenix

- Unicode-DFS-2016

- SWL

- Spencer-99

- Plexus

- VSL-1.0

- Leptonica

- Unicode-DFS-2015

- Mup

- Giftware

- OLDAP-2.2

- APAFML

- NBPL-1.0

- OLDAP-1.1

- Entessa

- Multics

- Newsletr

- psutils

- bzip2-1.0.5

- Afmparse

- diffmark

- BSD-2-Clause-Views

- DSDP

- MIT-Modern-Variant

- ANTLR-PD-fallback

- Bahyph

- BSD-3-Clause-Modification

- BSD-4-Clause-Shortened

- HTMLTIDY

- MIT-open-group

- MulanPSL-2.0

- OLDAP-2.0.1

- Saxpath

- Borceux

- Crossword

- CrystalStacker

- Rdisc

- Wsuipa

## C   Excluded file extensions

Part of this list was taken from `https://github.com/EleutherAI/github-downloader/blob/345e7c4cbb9e0dc8a0615fd995a08bf9d73b3fe6/download_repo_text.py`

'apk', 'app', 'bin', 'bmp', 'bz2', 'class', 'csv', 'dat', 'db', 'deb', 'dll', 'dylib', 'egg', 'eot', 'exe', 'gif', 'gitignore', 'glif', 'gradle', 'gz', 'ico', 'jar', 'jpeg', 'jpg', 'lib', 'lo', 'lock', 'log', 'mp3', 'mp4', 'nar', 'o', 'ogg', 'otf', 'p', 'pdb', 'pdf', 'png', 'pickle', 'pkl', 'ppt', 'pptx', 'pyc', 'pyd', 'pyo', 'rar', 'rkt', 'so', 'ss', 'svg', 'tar', 'tif', 'tiff', 'tsv', 'ttf', 'war', 'wav', 'webm', 'woff', 'woff2', 'xz', 'zip', 'zst'

## D   Included programming language extensions

This list of programming language extensions is taken from `https://gist.github.com/ppisarczyk/43962d06686722d26d176fad46879d41`.

.abap .asc .ash .ampl .mod .g4 .apib .apl .dyalog .asp .asax .ascx .ashx .asmx .aspx .axd .dats .hats .sats .as .adb .ada .ads .agda .als .apacheconf .vhost .cls .applescript .scpt .arc .ino .asciidoc .adoc .asc .aj .asm .a51 .inc .nasm .aug .ahk .ahkl .au3 .awk .auk .gawk .mawk .nawk .bat .cmd .befunge .bison .bb .bb .decls .bmx .bsv .boo .b .bf .brs .bro .c .cats .h .idc .w .cs .cake .cshtml .csx .cpp .c++ .cc .cp .cxx .h .h++ .hh .hpp .hxx .inc .inl .ipp .tcc .tpp .c-objdump .chs .clp .cmake .cmake.in .cob .cbl .ccp .cobol .cpy .css .csv .capnp .mss .ceylon .chpl .ch .ck .cirru .clw .icl .dcl .click .clj .boot .cl2 .cljc .cljs .cljs.hl .cljscm .cljx .hic .coffee ._coffee .cake .cjsx .cson .iced .cfm .cfml .cfc .lisp .asd .cl .l .lsp .ny .podsl .sexp .cp .cps .cl .coq .v .cppobjdump .c++-objdump .c++objdump .cpp-objdump .cxx-objdump .creole .cr .feature .cu .cuh .cy .pyx .pxd .pxi .d .di .d-objdump .com .dm .zone .arpa .d .darcspatch .dpatch .dart .diff .patch .dockerfile .djs .dylan .dyl .intr .lid .E .ecl .eclxml .ecl .sch .brd .epj .e .ex .exs .elm .el .emacs .emacs.desktop .em .emberscript .erl .es .escript .hrl .xrl .yrl .fs .fsi .fsx .fx .flux .f90 .f .f03 .f08 .f77 .f95 .for .fpp .factor .fy .fancypack .fan .fs .for .eam.fs .fth .4th .f .for .forth .fr .frt .fs .ftl .fr .g .gco .gcode .gms .g .gap .gd .gi .tst .s .ms .gd .glsl .fp .frag .frg .fs .fsh .fshader .geo .geom .glslv .gshader .shader .vert .vrx .vsh .vshader .gml .kid .ebuild .eclass .po .pot .glf .gp .gnu .gnuplot .plot .plt .go .golo .gs .gst .gsx .vark .grace .gradle .gf .gml .graphql .dot .gv .man .1 .1in .1m .1x .2 .3 .3in .3m .3qt .3x .4 .5 .6 .7 .8 .9 .l .me .ms .n .rno .roff .groovy .grt .gtpl .gvy .gsp .hcl .tf .hlsl .fx .fxh .hlsli .html .htm .html.hl .inc .st .xht .xhtml .mustache .jinja .eex .erb .erb.deface .phtml .http .hh .php .haml .haml.deface .handlebars .hbs .hb .hs .hsc .hx .hxsl .hy .bf .pro .dlm .ipf .ini .cfg .prefs .pro .properties .irclog .weechatlog .idr .lidr .ni .i7x

.iss .io .ik .thy .ijs .flex .jflex .json .geojson .lock .topojson .json5 .jsonld .jq .jsx .jade .j .java .jsp .js .__js .bones .es .es6 .frag .gs .jake .jsb .jscad .jsfl .jsm .jss .njs .pac .sjs .ssjs .sublime-build .sublime-commands .sublime-completions .sublime-keymap .sublime-macro .sublime-menu .sublime-mousemap .sublime-project .sublime-settings .sublime-theme .sublime-workspace .sublime_metrics .sublime_session .xsjs .xsjslib .jl .ipynb .krl .sch .brd .kicad_pcb .kit .kt .ktm .kts .lfe .ll .lol .lsl .lslp .lvproj .lasso .las .lasso8 .lasso9 .ldml .latte .lean .hlean .less .l .lex .ly .ily .b .m .ld .lds .mod .liquid .lagda .litcoffee .lhs .ls .__ls .xm .x .xi .lgt .logtalk .lookml .ls .lua .fcgi .nse .pd_lua .rbxs .wlua .mumps .m .m4 .m4 .ms .mcr .mtml .muf .m .mak .d .mk .mkfile .mako .mao .md .markdown .mkd .mkdn .mkdown .ron .mask .mathematica .cdf .m .ma .mt .nb .nbp .wl .wlt .matlab .m .maxpat .maxhelp .maxproj .mxt .pat .mediawiki .wiki .m .moo .metal .minid .druby .duby .mir .mirah .mo .mod .mms .mmk .monkey .moo .moon .myt .ncl .nl .nsi .nsh .n .axs .axi .axs.erb .axi.erb .nlogo .nl .lisp .lsp .nginxconf .vhost .nim .nimrod .ninja .nit .nix .nu .numpy .numpyw .numsc .ml .eliom .eliomi .ml4 .mli .mll .mly .objdump .m .h .mm .j .sj .omgrofl .opa .opal .cl .opencl .p .cls .scad .org .ox .oxh .oxo .oxygene .oz .pwn .inc .php .aw .ctp .fcgi .inc .php3 .php4 .php5 .phps .phpt .pls .pck .pkb .pks .plb .plsql .sql .sql .pov .inc .pan .psc .parrot .pasm .pir .pas .dfm .dpr .inc .lpr .pp .pl .al .cgi .fcgi .perl .ph .plx .pm .pod .psgi .t .6pl .6pm .nqp .p6 .p6l .p6m .pl .pl6 .pm .pm6 .t .pkl .l .pig .pike .pmod .pod .pogo .pony .ps .eps .ps1 .psd1 .psm1 .pde .pl .pro .prolog .yap .spin .proto .asc .pub .pp .pd .pb .pbi .purs .py .bzl .cgi .fcgi .gyp .lmi .pyde .pyp .pyt .pyw .rpy .tac .wsgi .xpy .pytb .qml .qbs .pro .pri .r .rd .rsx .raml .rdoc .rbbas .rbfrm .rbmnu .rbres .rbtbar .rbuistate .rhtml .rmd .rkt .rktd .rktl .scrbl .rl .raw .reb .r .r2 .r3 .rebol .red .reds .cw .rpy .rs .rsh .robot .rg .rb .builder .fcgi .gemspec .god .irbrc .jbuilder .mspec .pluginspec .podspec .rabl .rake .rbuild .rbw .rbx .ru .ruby .thor .watchr .rs .rs.in .sas .scss .smt2 .smt .sparql .rq .sqf .hqf .sql .cql .ddl .inc .prc .tab .udf .viw .sql .db2 .ston .svg .sage .sagews .sls .sass .scala .sbt .sc .scaml .scm .sld .sls .sps .ss .sci .sce .tst .self .sh .bash .bats .cgi .command .fcgi .ksh .sh.in .tmux .tool .zsh .sh-session .shen .sl .slim .smali .st .cs .tpl .sp .inc .sma .nut .stan .ML .fun .sig .sml .do .ado .doh .ihlp .mata .matah .sthlp .styl .sc .scd .swift .sv .svh .vh .toml .txl .tcl .adp .tm .tcsh .csh .tex .aux .bbx .bib .cbx .cls .dtx .ins .lbx .ltx .mkii .mkiv .mkvi .sty .toc .tea .t .txt .fr .nb .ncl .no .textile .thrift .t .tu .ttl .twig .ts .tsx .upc .anim .asset .mat .meta .prefab .unity .uno .uc .ur .urs .vcl .vhdl .vhd .vhf .vhi .vho .vhs .vht .vhw .vala .vapi .v .veo .vim .vb .bas .cls .frm .frx .vba .vbhtml .vbs .volt .vue .owl .webidl .x10 .xc .xml .ant .axml .ccxml .clixml .cproject .csl .csproj .ct .dita .ditamap .ditaval .dll.config .dotsettings .filters .fsproj .fxml .glade .gml .grxml .iml .ivy .jelly .jsproj .kml .launch .mdpolicy .mm .mod .mxml .nproj .nuspec .odd .osm .plist .pluginspec .props .ps1xml .psc1 .pt .rdf .rss .scxml .srdf .storyboard .stTheme .sublime-snippet .targets .tmCommand .tml .tmLanguage .tmPreferences .tmSnippet .tmTheme .ts .tsx .ui .urdf .ux .vbproj .vcxproj .vssettings .vxml .wsdl .wsf .wxi .wxl .wxs .x3d .xacro .xaml .xib .xlf .xliff .xmi .xml.dist .xproj .xsd .xul .zcml .xsp-config .xsp.metadata .xpl .xproc .xquery .xq .xql .xqm .xqy .xs .xslt .xsl .xojo_code .xojo_menu .xojo_report .xojo_script .xojo_toolbar .xojo_window .xtend .yml .reek .rviz .sublime-syntax .syntax .yaml .yaml-tmlanguage .yang .y .yacc .yy .zep .zimpl .zmpl .zpl .desktop .desktop.in .ec .eh .edn .fish .mu .nc .ooc .rst .rest .rest.txt .rst.txt .wisp .prg .ch .prw

