# OpenReview forum: "The Stack: 3 TB of permissively licensed source code"
_TMLR — Accepted by TMLR_

### Review · Reviewer_qZns · 2023-01-10

**Summary Of Contributions:**

The paper presents a dataset that consists of 3.1 TB of licensed source code in 30 programming languages.  The authors also manage to reproduce the results of Codex and CodeGen with only licensed code.

**Audience:**

Yes

**Broader Impact Concerns:**

concerns on the ethical implications of the work have been sufficiently addressed

**Claims And Evidence:**

No

**Requested Changes:**

See strengths and weaknesses section

**Strengths And Weaknesses:**

The paper proposes a dataset of licensed code which is clearly valuable to the community. However, I find the current version of the manuscript needs significant improvement before publication. See detailed comments below

- The paper needs to give a systematic comparison between The Stack and prior works [1,2,3] (which by the way are not even cited in the work) to illustrate the strengths and weaknesses of your datasets. \
[1] CodeXGLUE: A Machine Learning Benchmark Dataset for Code Understanding and Generation \
[2] CodeNet: A Large-Scale AI for Code Dataset for Learning a Diversity of Coding Tasks \
[3] COSET: A Benchmark for Evaluating Neural Program Embeddings

- In order to show the utility of the dataset, authors tried to reproduce the results of Codex and CodeGen. This is on the right track, unfortunately, it falls short on several fronts. First, why are only two models considered? Second, Why only Codex and CodeGen but not others? Third, why only consider the task of text2code. As a trademark of LLMs that are powered by few-shot learning paradigm (not the fine-tuning approach towards the specific downstream tasks), The Slack needs to enable such LLMs in a variety of code text related tasks.

- The authors need to make the link of the dataset public for evaluation. Since this is the probably the only contribution of the work, I don't believe the paper is acceptable without the open access to The Slack.

---

> ### Author Response · Authors · 2023-02-07
> **Response**
>
> Thank you for your review and constructive feedback! We respond to your specific concerns below.
>
> ### Re: Systematic comparison with prior works
> Thank you for bringing these prior works to our attention. We will incorporate them into the related work.
>
> ### Re: Other text2code models and other tasks
> In this work, we chose depth over breadth and carefully investigated pre-training on the Python subset. Prior work on code LLMs (Codex, CodeGen, InCoder) have consistently reported performance on python code completion benchmarks (HumanEval, MBPP) and we thought it would be most valuable to the community to first investigate what preprocessing pipeline can reproduce these prior results (especially since data processing details are not shared). We do agree that this work would be stronger if The Stack was evaluated on other programming languages. During this review process, there has been follow-up work (with a different author list) showing promising text2code results on other programming languages. Specifically, we trained 1.1B parameter models on the Java, JavaScript, and Python portions of The Stack and found the model to outperform prior open-source multilingual code generation models (InCoder-6.7B and CodeGen-Multi-2.7B) in both left-to-right generation and infilling, despite being substantially smaller. That work also reports encouraging results for docstring generation. To not break anonymity, we provide you a link to the anonymized version of this work: https://drive.google.com/file/d/1e4ZZoUJAIzrmuHbM6dU_xwx0s3WdBDAH/view?usp=share_link. We will mention and cite these results if the paper gets accepted.
>
> ### Re: link to the dataset
> The dataset is already widely available (e.g. on the HuggingFace hub) and downloaded over 10k times. We can’t share the link if we don’t want to break the double blind review process.

---

### Review · Reviewer_xNBc · 2023-01-22

**Summary Of Contributions:**

The authors compile a large dataset (called The Stack) of permissively-licensed open-source code, with the intention that the dataset can be used to train a language model. Unlike prior datasets of code, The Stack only includes code identified as having a permissive license such as the MIT or Apache 2.0 licenses, excluding code with a copyleft license; empowers developers to request removal of their code from the dataset; and is much larger than the earlier CodeParrot dataset. The authors also train some models on the Python subset of the dataset to evaluate how near-deduplication and using only the permissively-licensed data (versus all data) impact the results.

**Audience:**

Yes

**Broader Impact Concerns:**

Regarding the "Malicious code" paragraph of the Limitations section, I'm not sure if I agree that removing malicious code from the dataset necessarily would have positive social impact. For example, LMs which are aware of malicious code may be generally useful for analyzing and detecting malicious code, not just for generating new copies of it.

**Claims And Evidence:**

Yes

**Requested Changes:**

To secure recommendation:
- Please include an experiment or other empirical analysis which covers a part of the dataset other than only Python.

To strengthen the work:
- The end of page 7 states that 3 langauges only have less than 4% of the data marked with a permissive license. It would be nice if there was an augmented version of Table 3 to include this information for all languages.

**Strengths And Weaknesses:**

# Strengths
- The large dataset can be a useful resource for the research community.
- The dataset collection and curation process is designed to address various societal concerns that have been raised, such as copyright issues and allowing code authors to opt out of collection.
- The authors provide empirical evidence about the benefits of near-deuplication through their experiments.

# Weaknesses
- There is no empirical justification for how the precise near-deduplication method was selected, for example to justify the pipeline or the choice of 0.85 for Jaccard similarity.
- The fraction of data remaining after near-deduplication varies significantly depending on the language. It would be useful to have a study of why this might be, and if some of it is due to the language itself (e.g. a language requiring more common bolierplate than another) versus how people happen to use the language on GitHub.
- The experiments are only done using the Python subset of the dataset, and not any other languages.
- A significant fraction (about 1/4 before deduplication) of the dataset is HTML, for which we may expect that a significant fraction of the content is actually natural language, due to the nature of markup. As such, it may be that the HTML data is significantly different from the other programming language data.
- I think there would be no particular legal issues with the authors distributing copyleft data as part of the dataset, as long as it's clearly marked, and the downstream users can decide what to do with it. However, it looks like that part of the dataset is relatively small (<5%) anyways.

---

> ### Author Response · Authors · 2023-02-07
> **Response**
>
> Thank you for your thorough review and suggestion to improve our work! We address your points below.
>
> ### re: other programming languages than Python
> In this work, we chose depth over breadth and carefully investigated pre-training on the Python subset. Prior work on code LLMs (Codex, CodeGen, InCoder) have consistently reported performance on python code completion benchmarks (HumanEval, MBPP) and we thought it would be most valuable to the community to first investigate what preprocessing pipeline can reproduce these prior results (especially since data processing details are not shared). We do agree that this work would be stronger if The Stack was evaluated on other programming languages. During this review process, there has been follow-up work (with a different author list) showing promising text2code results on other programming languages. Specifically, we trained 1.1B parameter models on the Java, JavaScript, and Python portions of The Stack and found the model to outperform prior open-source multilingual code generation models (InCoder-6.7B and CodeGen-Multi-2.7B) in both left-to-right generation and infilling, despite being substantially smaller. That work also reports encouraging results for docstring generation. To not break anonymity, we provide you a link to the anonymized version of this work: [https://drive.google.com/file/d/1e4ZZoUJAIzrmuHbM6dU_xwx0s3WdBDAH/view?usp=share_link](https://drive.google.com/file/d/1e4ZZoUJAIzrmuHbM6dU_xwx0s3WdBDAH/view?usp=share_link). We will mention and cite these results if the paper gets accepted.
>
> ### re: add percentage of permissively licensed data to Table 3
> Thank you for this suggestion. We’ll add this information in the revision.
>
> ### re: justification near-deduplication pipeline
> We followed the hyperparameters of the deduplication pipeline of [1] and used the default Jaccard similarity (0.85) specified in their code base [2]. We will update the paper to include this information. Note that we explored the deduplication hyper parameters in more depth in the follow-up work we mentioned before: https://drive.google.com/file/d/1e4ZZoUJAIzrmuHbM6dU_xwx0s3WdBDAH/view?usp=share_link
>
> [1] The Adverse Effects of Code Duplication in Machine Learning Models of Code, Allamanis et al
> [2] https://gist.github.com/mallamanis/ce1a3624b6d1a9ec9b6966e6b7181dcd
>
> ### re: analysing the impact of near-deduplication for different programming languages
> That’s an interesting observation! We will provide some context/analysis after adding the percentages to Table 3.

---

### Review · Reviewer_tEET · 2023-01-30

**Summary Of Contributions:**

The paper presents a new dataset called "The Stack" to enable the learning of large language models (LLMs) for code. The dataset contains code written in 30 diverse programming languages. Overall, the dataset is larger than the existing ones. The authors implement near de-duplication in their collected dataset and describe methods to remove licensed code as much as possible. They also describe a mechanism through which developers can pull their data out of the stack. The authors evaluate the usefulness of their dataset on two challenging tasks: HumanEval and MBPP. They compare the performance of the trained models against models trained with state-of-the-art datasets. The results show that models trained with the stack can achieve better or comparable performance than existing methods.

**Audience:**

Yes

**Broader Impact Concerns:**

The authors have sufficiently addressed the potential negative impact of their proposed research in Section 6.

**Claims And Evidence:**

No

**Requested Changes:**

* Given that Python represents a small subset of all the collected code, I would recommend the authors show the suitability of the proposed dataset in other languages where they have more data as shown in Figure 1 (e.g., Javascript, HTML, or Java).

* I did not get the definition of the pass@k metric completely. You mention "k samples are generated for each problem, a problem is solved if at least one of the samples passes the test cases, and the total fraction of problem solved is measured", what is the meaning of samples here? input output example from the generated program? also as per the provided definition for pass@1, even if one sample out of 100 passes the test cases, the program is correct, isn't it wrong?

* In the paragraph on "How can I request that my data be removed from The Stack", what are the criteria you are going to apply for processing a user request? Only username, email address, and git activity need to be provided? How will you prevent developers from asking to remove permissive code from the database? More details here need to be provided.

* In Figure 2, it will help to specifically name the languages as you do in Figure 1.

**Strengths And Weaknesses:**

+ Stack has the potential to have a transformative impact on LLM for code by becoming the standard dataset for developing LLM for code models. In general, better datasets tend to lead to breakthroughs in AI.

+ The paper is in general well-written, easy to read, and accessible to a broad community.

+ The evaluation results on the Python language tasks show the promise of the proposed dataset.

- From a technical perspective, I did not see any particular algorithmic challenges solved here. The presentation in Section 3 describes relatively low-level implementation details (maybe there could have been some challenges in de-duplication but the authors rely on existing methods).

- Given that Stack supports code in 30 languages, I find it underwhelming that the authors chose to evaluate the quality of models trained with their dataset on only Python. Is there any good reason why other languages are not considered? Having these experiments will help the readers better judge the suitability of the proposed dataset for a variety of tasks.

---

> ### Author Response · Authors · 2023-02-07
> **Response to review**
>
> Thank you for your review! It’s encouraging that you see the potential transformative impact of The Stack as it could become the standard dataset for pre-training code LLMs.
>
> ### re: Evaluation on other programming languages than Python
> In this work, we chose depth over breadth and carefully investigated pre-training on the Python subset. Prior work on code LLMs (Codex, CodeGen, InCoder) have consistently reported performance on python code completion benchmarks (HumanEval, MBPP) and we thought it would be most valuable to the community to first investigate what preprocessing pipeline can reproduce these prior results (especially since data processing details are not shared). We do agree that this work would be stronger if The Stack was evaluated on other programming languages. During this review process, there has been follow-up work (with a different author list) showing promising text2code results on other programming languages. Specifically, we trained 1.1B parameter models on the Java, JavaScript, and Python portions of The Stack and found the model to outperform prior open-source multilingual code generation models (InCoder-6.7B and CodeGen-Multi-2.7B) in both left-to-right generation and infilling, despite being substantially smaller. That work also reports encouraging results for docstring generation. To not break anonymity, we provide you a link to the anonymized version of this work: [https://drive.google.com/file/d/1e4ZZoUJAIzrmuHbM6dU_xwx0s3WdBDAH/view?usp=share_link](https://drive.google.com/file/d/1e4ZZoUJAIzrmuHbM6dU_xwx0s3WdBDAH/view?usp=share_link). We will mention and cite these results if the paper gets accepted.
>
> ### re: pass@k rate
> This metric was introduced in the Codex paper [1] and we’ve adopted it here to compare with other models from the literature. Samples mean generated completions by the model for a given task prompt. For pass@100, the problem is considered solved if one of the 100 model samples passes all the provided unit tests. We will update the paper to make this more clear.
>
> [1] Evaluating Large Language Models Trained on Code, Chen et al.
>
> ### re: criteria opt-out requests
> The opt-out procedure is on top of the permissively licensed source code. Even if a developer’s repository is permissively licensed, we do want to give them the possibility to have their code removed from The Stack, as we acknowledge that the code creator might not have had the intention to have their data used for developing LLMs.
>
> ### re: fig. 2 legend
> Thank you for pointing this out. We’ll address this in the revision.

---

### Decision · Action_Editors · 2023-04-18

**Recommendation:** Accept as is

**Comment:**

As mentioned, the absence of results on languages
other than Python is a significant weakness. On balance, however, the
ML-for-code community can benefit from more large, open, permissively
licensed datasets, and this dataset appears to be a valuable
resource. Given this, I am recommending acceptance. Please incorporate
all reviewer feedback into the final version.

**Audience:**

The dataset will be a valuable resource for the growing -- and
increasingly mainstream -- machine-learning-for-code community.

**Claims And Evidence:**

The paper introduces a dataset, called the Stack, consisting of 3.1 TB
of permissively licensed code in 30 languages. The authors have
thought carefully about the governance of the dataset, for example, by
developing a tool to check if a developer's data is in the Stack and
by including opt-out provisions for developers. They have also
performed a reasonable dataset analysis and discussed the paper's
limitations in some depth. However, the experimental section is quite
weak -- results are only shown for the Python subset of the dataset.
The authors have written a different paper that uses the Java and
Javascript portions of the data, but the results from that paper
cannot be included here. Also, 350 M parameters are not that many in
2023.